



# *"Thanks for helping me find my enthusiasm for physics!"* The lasting impacts 'research in schools' projects can have on students, teachers, and schools

Martin O. Archer[1] and Jennifer DeWitt[2,3]

[1]School of Physics and Astronomy, Queen Mary University of London, London, UK
[2]Institute of Education, University College London, London, UK
[3]Independent Research and Evaluation Consultant, UK

**Correspondence:** Martin O. Archer
(martin@martinarcher.co.uk)

**Abstract.** Using 6 years of evaluation data we assess the medium- and long-term impacts upon a diverse range of students, teachers, and schools from participating in a programme of protracted university-mentored projects based in cutting-edge space science, astronomy, and particle physics research. After having completed their 6-month-long projects, the 14–18 year-old school students report having substantially increased in confidence relating to relevant scientific topics and methods as

well as having developed numerous skills, outcomes which are corroborated by teachers. There is evidence that the projects helped increase students' aspirations towards physics, whereas science aspirations (generally high to begin with) were typically maintained or confirmed through their involvement. Longitudinal evaluation 3 years later has revealed that these projects have been lasting experiences for students which they have benefited and drawn upon in their subsequent university education. Data on students' destinations suggests that their involvement in research projects has made them more likely to undertake physics

and STEM degrees than would otherwise be expected. Cases of co-created novel physics research resulting from PRiSE also has seemed to have a powerful effect, not only on the student co-authors but participating students from other schools also. Teachers have also been positively affected through participating, with the programme having influenced their own knowledge, skills, and pedagogy, as well as having advantageous effects felt across their wider schools. These impacts suggest that similar 'research in schools' initiatives may have a role to play in aiding the increased uptake and diversity of physics and/or STEM in

higher education as well as meaningfully enhancing the STEM environment within schools.

## 1 Introduction

Independent research projects provide extended opportunities for school students to lead and tackle open-ended scientific investigations, with so-called 'research in schools' programmes, which have been emerging in recent years, being a subset of these linked to current academic (STEM) research (e.g. Colle et al., 2007; M.O. Archer, 2017; Sousa-Silva et al., 2018; IRIS,

2020). It has been realised recently that, in general, more extended programmes of STEM interventions with young people are required to be effective in having lasting impact upon them compared to typical "one off" approaches (M.O. Archer et al.,





Under Review, and references therein). Independent research projects and 'research in schools', when appropriately developed and supported by expert mentors from universities or industry, thus potentially align with this direction.

Bennett et al. (2016, 2018) reviewed the evidence of impact from independent research project endeavours globally. They
found that impacts on students were most often investigated, with various outcomes being reported including improved understanding, practical and transferable skills, or attitudes and aspirations towards science. Dunlop et al. (2019) further suggest there is value in students participating in independent research projects through developing their understanding about scientific research/researchers and allowing them to make more informed decisions about their future subject choices. It appears at present though that only a few programmes consider the impact on students from under-represented groups, presenting emerg-
ing evidence that increased engagement with science can result from their involvement (Bennett et al., 2016, 2018). Finally, the review highlights that evaluations exploring the potential long-term impacts of project work on students, such as subsequent subject or career choices, are currently lacking.

While some studies into independent research projects use views from teachers, these tend to little explore the impact on the teachers themselves from their own participation in projects (Bennett et al., 2016, 2018). This aspect has recently been
considered by Rushton and Reiss (2019) for teachers engaged in projects with IRIS (2020). Through in-depth interviews with 17 research-active science teachers, they report that through the projects they felt reconnected with science or research and had developed as teachers, including in their pedagogy, skills development, and recognition by school colleagues.

In general, however, Bennett et al. (2016, 2018) note that with little detail on the assessment criteria for independent research projects being reported, further work is required to improve the quality of evidence and to more fully explore the potential long-
term benefits of teachers' and students' (particularly those from under-represented groups) involvement in independent research projects. This paper draws on 6 years' worth of evaluation data from the 'Physics Research in School Environments' (PRiSE) programme of independent research projects to explore the possible impacts on participating students, teachers and schools.

## 2   PRiSE

'Physics Research in School Environments' (PRiSE) is a scalable framework for independent research projects based in cur-
rent physics research that are mentored by active researchers (M.O. Archer et al., 2020). The programme aims to equip 14–18 year-old school students (particularly those from disadvantaged backgrounds) with the ability, confidence, and skills in order to increase/sustain their aspirations towards physics or more broadly STEM, ultimately enabling them to realise these at higher education and thus contributing to increased uptake and diversity of physics, and to some extent STEM (cf. L. Archer et al., 2020a). Through working with teachers, PRiSE also aims to develop their professional practice and build long-term
university–school relationships that raise the profile of science and mitigate biases/stereotypes associated with physics within these schools, generally making them environments which nurture and enhance all students' science capital (cf. IOP, 2014).

PRiSE was developed at Queen Mary University of London (QMUL) in 2014, where the four projects summarised in Table 1 currently exist. These projects run for approximately six months from the start of the UK academic year in September to just before the spring/Easter break in March, culminating with students presenting their work at a special conference held on



| Project | Abbreviation | Years | Field | Description |
|---|---|---|---|---|
| Scintillator Cosmic Ray Experiments into Atmospheric Muons | SCREAM | 2014–2020 | Cosmic Rays | Scintillator – Photomuliplier Tube detector usage |
| Magnetospheric Undulations Sonified Incorporating Citizen Scientists | MUSICS | 2015–2020 | Magnetospheric Physics | Listening to ultra-low frequency waves and analysing in audio software |
| Planet Hunting with Python | PHwP | 2016–2020 | Exoplanetary Transits | Learning computer programming, applying this to NASA Kepler and TESS data |
| ATLAS Open Data | ATLAS | 2017–2020 | Particle Physics | Interacting through online tool with LHC statistical data on particle collisions |

**Table 1.** A summary of the existing PRiSE projects at QMUL.

university campus. Students typically spend 1–2 hours per week throughout working on the project in groups, usually outside of lesson time. Support from the university is provided through workshops, school visits, monthly webinars, printed/multimedia resources, and ad hoc emails as required. The provision within the programme is explored in more detail in M.O. Archer et al. (2020). PRiSE has engaged a much more diverse set of school students and significantly more disadvantaged groups than is typical. Furthermore it has been found that students' success within the programme appears independent of background, which

has been attributed by teachers as due to the extraordinary level of support offered (M.O. Archer, 2020).

The evaluation of PRiSE's pilot, which ran from 2014–2016 and involved 6 schools, suggested that students' awareness of current scientific research, understanding of the scientific method, and skills were enhanced by the programme and that teachers benefited through reconnecting with their subject at an academic level, being challenged, and being supported in their professional development (M.O. Archer, 2017). The programme has grown significantly since then, having involved 67 London

schools by 2020. This paper expands the evaluation of PRiSE's potential impacts.

## 3   Methods

To evaluate the impact of the PRiSE programme, questionnaires were distributed to students and their teachers at each year's student conference held approximately 6 months after they started their PRiSE projects. We were also able to contact a subset of individuals three years after their participation.

### 3.1   Instruments

Paper questionnaires were handed out to students and teachers at our student conferences each year (apart from 2020 where this was done online due to the COVID-19 pandemic), which assessed the impact on students and teachers at this 6-month stage. There were two questionnaires for students. One of these was project-specific, relating to students' confidence in scientific topics/practices relevant to their specific project, i.e. those in Table 1. The other applied to PRiSE in general, asking

students about skills development, aspirations, and other ways they may have been affected by their involvement. Teachers





only completed a PRiSE-wide questionnaire, which not only asked for their observations of impacts upon students but also how involvement in the project has affected their own knowledge, skills, practice, and wider school environments. The questions posed to both students and teachers varied slightly from year-to-year and are found in Appendix A. The PRiSE-wide questionnaires also included feedback on participants' experience of the programme, with this data forming the focus of a
separate paper (M.O. Archer et al., 2020).

These instruments were chosen in order to collect data from as wide a range of students and teachers as possible as well as respecting the limited time/resources of all involved (both on the school and university sides). All data gathered was anonymous, with students and teachers only indicating their school and which project they were involved with. We have further anonymised the data by using pseudonyms for the schools. More detailed information about the schools involved can be found
in M.O. Archer (2020). No protected characteristics (such as gender or race) or sensitive information (such as socio-economic background) were recorded. Our ethics statement on the forms informed participants that the data was being collected for evaluation purposes to determine the programme's impact, and that they could leave any question they felt uncomfortable answering blank (also true of the online form).

For longer term evaluation, students were also asked on a separate paper form at our conference to share their personal
email addresses so that we could follow up with them a few years later, in order to explore potential lasting impacts of the programme. As with the main questionnaires, this was presented as optional with an ethics statement and description on how their data would be used clearly presented. It was decided to contact cohorts of PRiSE students 3 years after they started the project for this follow-up, so that students would either be studying at university or at least (in the case of the youngest PRiSE students) making university applications, hence giving us insight into university destinations/plans. The students were emailed
and asked to fill out an online form, detailed in Appendix B. The form contained primarily open-ended questions, to enable us to understand PRiSE students' long-term attitudes to the programme, their higher education destinations, and what may have affected these decisions. This sort of contextual data would not be available by simply obtaining destination data from services such as the Higher Education Access Tracker (HEAT, https://heat.ac.uk) or requesting that schools provide it as a condition of their participation. Doing so would have also risked schools declining to participate. However, we do acknowledge that our
approach reduces the number of responses that could realistically be collected.

## 3.2 Participants

Data was collected from 153 students (aged 14–18) and 45 teachers across 37 London schools. A breakdown of the number of responses per year and how many schools these responses came from is given in Table 2 where the total number of participants (and their schools) in attendance at our conferences are also indicated. We note that, due to the COVID-19 pandemic, the
2019/20 programme was disrupted so we do not have reliable information on how many students, teachers, and schools would have successfully completed the programme that year. There is no indication that the respondents differed in any substantive way from the wider cohorts participating in the programme.





|  |  | 2015 | 2016 | 2017 | 2018 | 2019 | 2020 |
|---|---|---|---|---|---|---|---|
| **Students** | Project-specific |  | 13/26 (50%) | 30/70 (43%) | 45/92 (49%) | 40/97 (41%) |  |
|  | PRiSE-wide |  | 13/26 (50%) | 21/70 (30%) | 46/92 (50%) | 38/97 (39%) | 35/? |
| **Teachers** | PRiSE-wide | 1/1 (100%) | 6/6 (100%) | 6/11 (55%) | 9/16 (56%) | 6/16 (38%) | 17/? |
| **Schools** |  | 1/1 (100%) | 6/6 (100%) | 11/ 11 (100%) | 13/15 (87%) | 11/15 (73%) | 19/? |

**Table 2.** Response rates to questionnaires at PRiSE student conferences.

## 3.3 Analysis

Both qualitative and quantitative approaches were utilised in data analysis, as the open and closed ended questions in the
surveys generated different types of data.

For all quantitative (numerical) data, uncertainties presented represent standard (i.e. 68%) confidence intervals. For pro-
portions/probabilities these are determined through the Clopper and Pearson (1934) method, a conservative estimate based
on the exact expression for the binomial distribution, and therefore represent the expected variance due to counting statistics
only. Several statistical hypothesis tests are used with effect sizes and two-tailed $p$-values being quoted, with a statistically
significant result being deemed as $p < 0.05$. In general we opt to use nonparametric tests as these are more conservative and
suffer from fewer assumptions (e.g. normality, interval-scaling) than their parametric equivalents such as t-tests (Hollander and
Wolfe, 1999; Gibbons and Chakraborti, 2011). When comparing unpaired samples a Wilcoxon rank-sum test is used, which
tests whether one sample is stochastically greater than the other (often interpreted as a difference in medians). The Wilcoxon
signed-rank test is used to compare both a single sample to a hypothetical value or data from paired samples to one another.
Both versions test whether differences in the data are symmetric about zero in rank. Finally, for proportions we use a binomial
test, an exact test based on the binomial distribution of whether a sample proportion is different from a hypothesized value
(Howell, 2007). For ease of reference, further details about the quantitative analyses are incorporated into the relevant sections
of the findings.

Thematic analysis (Braun and Clarke, 2006) was used to analyse the textual (qualitative) responses. Instead of using pre-
determined qualitative codes to categorise the data, our analyses drew on a grounded theory approach (Robson, 2011; Silver-
man, 2010), letting the themes emerge from the data itself. This process involved the following steps:

1. Familiarisation: Responses are read and initial thoughts noted.

2. Induction: Initial codes are generated based on review of the data.

3. Thematic Review: Codes are used to generate broad themes (which we refer to as dimensions) and identify associated
   data.

4. Application: Codes are reviewed through application to the full data set.

5. Reliability: Codes are applied to a subset of data by second coder to check reliability.



6. Final Coding: Final codes are applied to the data.

7. Analysis: Thematic overview of the data is confirmed, with examples chosen from the data to illustrate the themes
(dimensions).

Our analyses examine the impact on students and teachers from schools which completed the programme, as it was not possible to gather evidence from schools which dropped out during the year. Future work could attempt to investigate this, subject to funding and ethical approval.

## 4   Impact on students

This first section of the findings examines the impact of PRiSE on students at the 6-month (captured at our student conferences) and 3-year (captured online) stages. Impact on students through the co-production of research is also briefly discussed.

### 4.1   6-month stage evaluation

We assess the impact on students in three broad areas related to the aims of the programme: their confidence in scientific topics and methods; their skillsets; and their aspirations towards pursuing physics or STEM.

**4.1.1   Confidence**

At our conferences from 2016–2019 students ($n = 127$) were asked to rate their confidence (using a 6-point Likert scale) in topics and practices relevant to their projects (see Table 3). Additionally they were asked to reassess their confidence before having undertaken the project. Given that the options varied by project, this was performed on separate paper questionnaires to the more general PRiSE-wide questionnaire that applied to all projects, however, the same level of anonymity and ethical
considerations were afforded to students here too. While for a subset of students M.O. Archer (2017) also surveyed students before undertaking projects, we opted not to continue this. This is firstly because surveys taken on different days necessarily include natural intraindividual variability (Eid and Diener, 1999) which cannot be separated out from any real stable change with only two survey points. Secondly, a reassessment following the project also means that an apparent decrease in confidence, known as the Kruger and Dunning (1999) effect, is avoided because prior to the interventions confidence may have been
artificially high as individuals are not aware of what they do not know. Given that many other repeated STEM intervention programmes have resulted in no overall changes from before to after (e.g. Jeavans and Jenkins, 2017; Hope-Stone Research, 2018), we set the benchmark to be a statistically significant positive change.

We test the paired before and after data for each topic and practice, omitting any where students listed either as unsure, using a Wilcoxon signed-rank test. The results show statistically significant increases for all the topics and practices, with the range of
two-tailed $z$-scores by project listed in square brackets in Figure 1 ($z \geq 1.96$ corresponds to 95% confidence or more). To give an overall measure of these changes, we take an average for each student across all topics and practices (again omitting unsures as before) and these are plotted in Figure 1 showing $96 \pm 2\%$ of students have increased in confidence. The median overall



| | Topics | | | Practices |
| --- | --- | --- | --- | --- |
| SCREAM | MUSICS | PHwP | ATLAS | |
| Neutrinos | Plasma | Planets | Fundamental Particles | Mathematical Models |
| Muons | Magnetic Fields | Stars | Fundamental Forces | Experiment Design |
| Cosmic Rays | Space | Gravity | Particle Detectors | Calibration (SCREAM) |
| Particle Detectors | Magnetosphere | Exoplanet Detection | Particle Interactions | Statistical Analysis |
| Anti-particles | Waves | | | Error Analysis |
| Special Relativity | Resonance | | | Drawing Conclusions |
| | | | | Collaborating |
| | | | | Presenting |
| | | | | Writing |
| | | | | Reviewing Literature |
| | | | | Programming (PHwP) |

**Table 3.** The scientific topics and practices used in assessing students' confidence.

change across all projects was $0.92 \pm 0.04$ points, indicated as the black bar in Figure 1 along with interquartile range (grey area), whereas the mean was slightly higher at $1.08 \pm 0.06$ due to a positive skewness (the uncertainty refers to the standard error in the mean). The overall results show positive changes across all projects to a high level of confidence, as indicated in the figure, with no real variation in results between the different projects or between schools. Therefore, students' confidence in scientific topics and methods seems to have substantially increased as a result of PRiSE and almost all students reported this benefit. This gain in confidence has also been noted in teachers' comments:

> "*They have become more confident in communicating their ideas and realised that they are not too young to do research.*" (Teacher 1, Hogwarts, SCREAM 2015)
> "*This has been a challenging experience for the students taking part. Students have gained a better appreciation of real science and built confidence.*" (Teacher 3, Xavier's Institute for Higher Learning, MUSICS 2016)

### 4.1.2 Skills

From 2016 onwards we asked students ($n = 140$) to list which skills, if any, they felt they had developed through their PRiSE project. Teachers ($n = 40$) were also asked to indicate their observations on skills that their students had developed during the programme. We extracted keywords from any prose responses and sifted through the data performing keyword clustering. This latter step involved identifying synonymous skills and relabelling them so there was a consistency of terminology throughout. This processing resulted in a dataset of 79 unique skills. Students tended to identify on average around 2 distinct skills each whereas teachers typically listed 3, though the responses per person ranged up to 5 and 6 respectively. Figure 2 shows the skills identified as a word cloud, where students and teachers have been given equal weight by normalising their counts by their respective totals. Colours indicate from whom the words originated, showing a large amount of agreement between





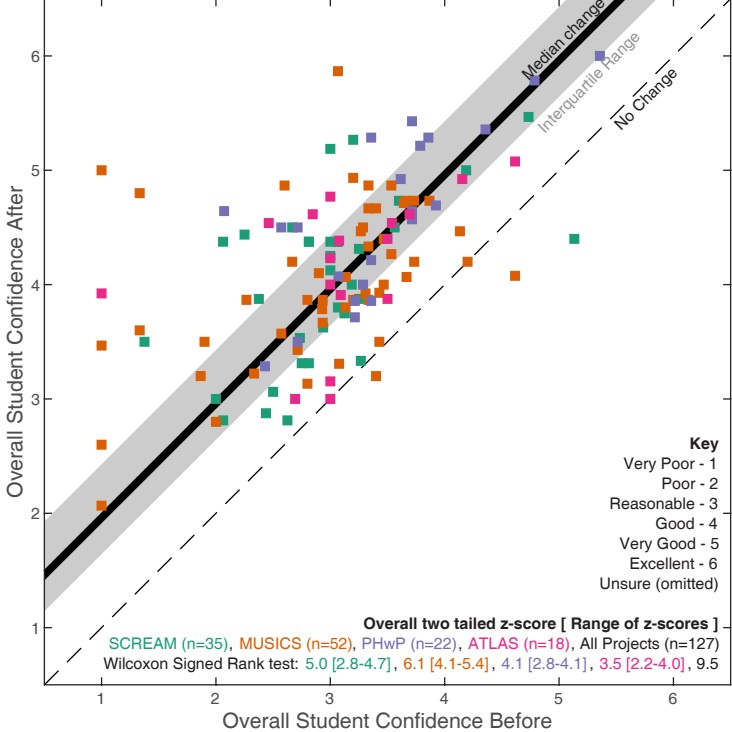

**Figure 1.** Overall student confidence in relevant scientific topics/methods before and after taking part in PRiSE ($n = 127$). Data points are coloured by project. Overall changes are also indicated through the interquartile range (grey area) and standard confidence interval in median (black bar). Overall and ranges of $z$-scores in two-tailed Wilcoxon signed-rank tests are also listed by project.

the two groups. All the skills listed are highly relevant to being a scientist, with the most cited being (in descending order) teamwork, research, data analysis, programming and presentation skills. These results remain fairly consistent with those from the pilot and arguably indicate areas where university / research physics differ substantially from the regular school experience.

Therefore, through experiencing and being involved in research-level physics, it seems likely students have gained new, or further developed existing, skills, constituting a positive impact upon them. This has been further expanded upon in the teacher feedback:

> "*They have developed presenting skills, they do get that* [at school] *but not for academic poster sessions. The unique skills from the project were the exposure to the physics, analysis, independence; it has allowed them to*
*access the world.*" (Teacher 1, Hogwarts, SCREAM 2015)
> "*Challenging opportunity* [for students] *to broaden skills and experience.*" (Teacher 23, Smeltings, ATLAS 2019)
> "*Great for developing pupil research skills and getting confident in cross referencing scientific articles, a very important skill for them in post-college education.*" (Teacher 42, Hill Valley High School, ATLAS 2020)



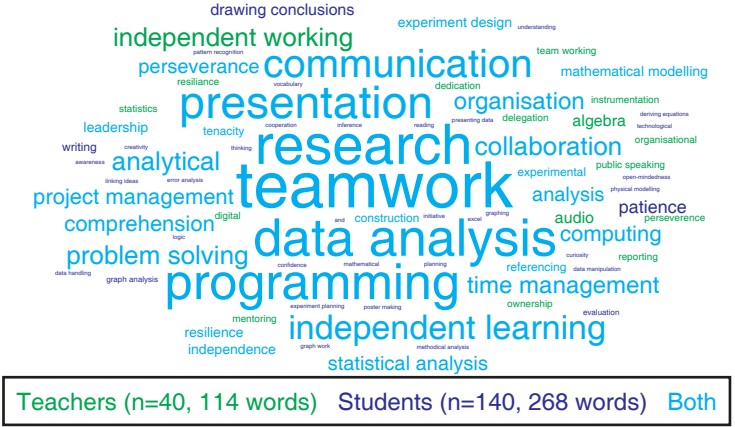

**Figure 2.** Word cloud of skills developed by students. Colours indicate words identified by students (blue), teachers (green), or both (cyan). Students and teachers have been given equal total weight.

### 4.1.3 Aspirations

To assess whether students' aspirations were affected at the 6-month stage, we first undertook a qualitative analysis in 2018–2019 asking in an open question how students' thoughts about future subject choices or careers might have been affected through doing the project. A thematic analysis of the 63 responses revealed three distinct dimensions, each with their own set of underlying codes. The first of these concerned how much the students' felt that they had wanted to study either physics or a STEM subject before even undertaking the project ($n = 22$), which tended to be raised if they already wanted to pursue this

route or if they were not interested in these subjects. The second dimension covers the students' aspirations following the project ($n = 13$), revealing that students were either now wanting to pursue physics/STEM, were considering these as potential options, or were simply unsure. Finally, the third dimension was an expression of change as a result of their involvement in PRiSE ($n = 22$), which typically stated that it had confirmed their subject choice, made them more likely to pursue physics/STEM, had not affected them, or (in a small number of cases) had deterred them from continuing with physics/STEM. These themes

align with the findings of Dunlop et al. (2019) on how independent research projects can affect students' aspirations.

We show the dimensions and codes in Table 4, also giving counts of the number of responses which fall within them (cf. Sandelowski, 2001; Sandelowski et al., 2009; Maxwell, 2010). We note that some students' responses covered more than one of the dimensions, but none spanned all three. Out of the 63 responses to this question, 11 did not fit into any of these three dimensions, instead highlighting aspects of the programme they enjoyed (research, teamwork, real applications of physics, and

what physics at university is like) but not explicitly stating their subject aspirations or how they may have been changed by the project. From these counts, it is clear that in both dimensions 2 and 3 the totals indicating positive effects from PRiSE are greater than the neutral or negative responses. However, given that these numbers are rather small and derived from a qualitative coding we do not attempt to make a statistical interpretation about the impact of PRiSE on students' physics/STEM aspirations based on them.





**Dimension 1: Likeliness of wanting to study physics/STEM before the project**

| Codes | Wanted to already | Unrelated to future choices |
|---|---|---|
| Count | 16 | 6 |

**Dimension 2: Likeliness of wanting to study physics/STEM after doing project**

| Codes | Now wants to | Now considering as option | Unsure on future choices |
|---|---|---|---|
| Count | 4 | 3 | 6 |

**Dimension 3: Change in likeliness of wanting to study physics/STEM due to project**

| Codes | Confirmed as subject choice | More likely | Didn't change mind | Deterred |
|---|---|---|---|---|
| Count | 6 | 6 | 8 | 2 |

**Table 4.** Qualitative coding of students' responses concerning their future aspirations along with counts ($n = 63$).

Instead, informed by these promising preliminary results, we implemented in 2020 a quantitative approach to assessing how PRiSE may have affected students' aspirations. In a similar manner and with similar justification to our evaluation of students' confidence, we asked students ($n = 35$) to assess their likelihood (using a 5-point Likert scale) of continuing with physics and STEM as well as reassessing these from before the project. We shall call this the 'absolute scale', since it pertains to students' absolute likelihood of continuing these subjects, and code it to values of 1–5 as shown in Figure 3a–b. In addition, we asked how working on the project affected their thoughts on physics and STEM as future subject choices using a different 5-point scale, where the wording of the options used were informed by the previous qualitative results in Table 4. We refer to this as the 'relative scale', since it concerns whether the students feel their thoughts changed as a result of PRiSE, and code it to values from -2 to +2, as shown in Figure 3c. One student only answered these questions relating to physics but not for STEM.

When considering physics aspirations, there was no clear positive bias towards the subject beforehand (see horizontal distribution in Figure 3a) with a mean value on the absolute scale of $3.23 \pm 0.21$ ($p = 0.264$ in a one-sample Wilcoxon signed-rank test against null hypothesis of 3). The vertical distribution shows some shift towards greater values on the absolute scale after the projects, now exhibiting overall positive results (mean of $3.69 \pm 0.20$, $p = 0.006$). From the paired data, $40 \pm 10\%$ of students increased in likelihood of studying physics on the absolute scale (though no students who were very unlikely before showed any increase) and only one student's likelihood decreased (to a neutral stance), with the mean change being $+0.46 \pm 0.12$. While this indicates only moderate changes in students' absolute physics aspirations, they are statistically significant ($p = 8 \times 10^{-4}$ in a Wilcoxon signed-rank test). On the relative scale displayed in the top panel of Figure 3c, however, $69 \pm 9\%$ of students report that the projects either made them more likely to continue with physics or confirmed it, with again only one student becaming less likely to pursue physics. The average was $+0.89 \pm 0.13$, greater than zero with high confidence ($p = 2 \times 10^{-5}$). No trends were present by project or school. Since we colour the datapoints in panel a of Figure 3 based on the students' responses on the





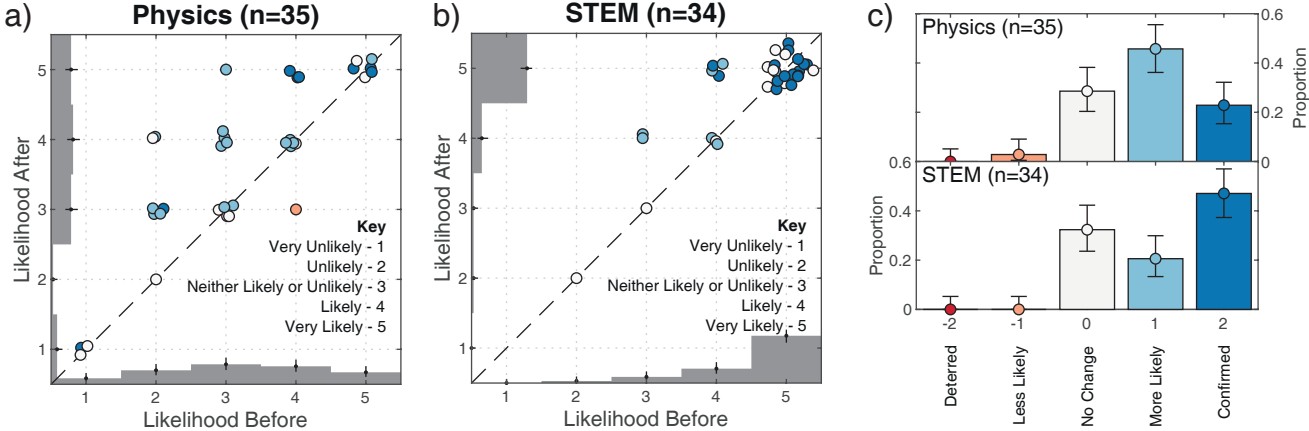

**Figure 3.** Likelihood before and after of PRiSE students continuing with a) physics and b) STEM education. Datapoints have been jittered for visibility. Marginal distributions (grey areas) are also shown for both before and after. How PRiSE specifically has affected students' aspirations are shown in panel c for physics (top) and STEM (bottom), where the colours used are also reflected in panels a and b. Error bars denote the standard Clopper and Pearson (1934) interval.

relative scale (panel c), it is clear that most of the students whose absolute physics aspirations increased attribute this in some way to PRiSE, while around half of those which did not increase on the absolute scale still claim positive influence by PRiSE.

Students' STEM aspirations, unlike physics, were already incredibly high before the projects as shown in Figure 3b with a mean value on the absolute scale of $4.53 \pm 0.14$. Because of this, a large proportion ($67 \pm 9\%$) of students would be unable to increase in value on the absolute scale due to already giving the highest rating. No students decreased in likelihood on the

absolute scale and only 6 students out of the 34 increased, though this constituted around half of the students who could possibly increase (gave a 4 or below beforehand). This slight change in the paired data (average $+0.18 \pm 0.07$ across all students) was still statistically significant ($p = 0.031$). The bottom panel of Figure 3c reveals a bimodal distribution on the relative scale of whether PRiSE affected students' STEM aspirations, with the largest peak at +2 (it confirmed their subject choice) and a smaller one at 0 (no change). No students reported being less likely to pursue STEM due to PRiSE. A similar proportion to with

physics of $68 \pm 9\%$ indicated PRiSE's likely positive influence on their STEM aspirations, with the mean being $1.15 \pm 0.15$ which is again a clear positive result ($p = 1 \times 10^{-5}$). As before there was no real variation in these results by project or school. All students who increased on the absolute scale attribute this to PRiSE, whereas a majority who did not change still indicate PRiSE had a positive effect on their STEM aspirations.

Ideally one would benchmark the likelihoods before PRiSE against larger surveys of similarly-aged students' aspirations

to test whether PRiSE students were more likely to continue with physics/STEM anyway. Unfortunately, direct comparisons are not possible due to the differing ways the relevant questions have been structured across national surveys. However, such research has shown that STEM degree aspirations amongst all students remains similar to the makeup of STEM vs. non-STEM A-Level subject choices, implying that almost all students studying at least one STEM A-Level likely aspire towards a STEM





degree (Hamlyn et al., 2020). Furthermore, science aspirations are highly correlated to 'science capital' and begin to form at an early age (Moote et al., 2020). Therefore, it is not surprising that PRiSE students' likelihood of wanting to continue with STEM was high beforehand.

The follow up qualitative question, asking students to explain how or why their thoughts about subject choices had been affected by the project typically mentioned how their interest, enjoyment, or understanding had been enhanced, e.g.

*"Before working on Planet Hunting With Python, I was already quite focused on studying in a STEM field, and the main reason I signed up for the project was because of my interest in physics. After the project, I felt as though my decision to pursue such an area was only further cemented."* (Student 120, Octavian Country Day School, PHwP 2020)

The one student who reported being less likely to pursue physics (but was not affected with regards to STEM) noted that

*"I already had my mind set on doing STEM subjects at university, but now I am less interested in physics as I have come to see how some areas are more challenging than others and I wouldn't want to specialise in those areas."* (Student 136, Jedi Academy, ATLAS 2020)

In light of this response, arguably the project enabled this student to make a more informed decision (cf. Dunlop et al., 2019) rather than necessarily causing harm. Some students raised the idea of their science identity affecting their likelihood beforehand, a known key factor in students' aspirations (e.g. L. Archer et al., 2013, 2020b), with PRiSE showing mixed results in affecting this

*"I never really saw Physics as a choice for me, as I did not want to do it, and the project hasn't changed my mind about this."* (Student 133, Imperial Academy, ATLAS 2020)
*"I am not very great at physics but this project made me more interested and invested."* (Student 135, Xavier's Institute for Higher Learning, SCREAM 2020)

Another theme that emerged in response to this question was that PRiSE added something not usually accessible to them in school, the research, which helped cement students' subject choices and potentially influenced their career aspirations (cf. L. Archer et al., 2020a)

*"Because I've always wanted to go into research and this project showed me that I enjoy it."* (Student 138, Jedi Academy, ATLAS 2020)
*"I've always assumed that I would work towards studying physics since GCSE; this project was very useful in seeing what research may be like if I took that path after education."* (Student 143, Sunnydale High School, MUSICS 2020)
*"I was always interested in maths and physics, especially maths. However this project showed me what we do not do in physics lessons, the research. For me this is one of the most important things in physics."* (Student 145, Sunnydale High School, MUSICS 2020)





Students' aspirations have been found to be extremely resilient and very difficult to change, with most (even protracted series of) interventions yielding no statistically significant overall effects (L. Archer et al., 2013, 2014; M.O. Archer et al., Under Review). With this in mind, the moderate increase in physics and the slight positive change in STEM absolute aspirations are considerable compared to the sector and what is realistically achievable with the resources afforded. The relative scale indicates

that students feel that PRiSE has influenced their subject choices, typically either confirming or making them more likely to follow either physics or STEM, with $80 \pm 8\%$ of students indicating a positive effect in either or both.

### 4.2   3-year stage evaluation

To date we have undertaken long-term evaluation for three cohorts of PRiSE students who had participated in the academic years 2015/16 (cohort 1), 2016/17 (cohort 2), and 2017/18 (cohort 3). At our student conferences 72 students from these three

cohorts left contact details with us for this purpose, which were well spread across the different schools involved. Across the three cohorts, the bounce rate was $46 \pm 7\%$ (predominantly due to now inactive school email addresses being given) and the emails were opened by 24 PRiSE students (a rate of $62 \pm 9\%$ from non-bouncing emails, again well spread across different schools) with 14 filling out the survey though we purposely cannot identify individuals from responses. While this is a relatively small number of responses, longitudinal evaluation is notoriously difficult for under-resourced engagement programmes (e.g.

M.O. Archer et al., Under Review) and there are still significant and useful results from the data, which we present in this section. The evaluation covers the perceived legacy of PRiSE on these students, as well as the students' higher education destinations and the factors which may have affected these.

#### 4.2.1   Legacy

Most of the 14 PRiSE students who responded were aged 16–17 when they undertook their projects, with two students aged 15–

16, and one student each in the ranges 14–15 and 17–18. All of them remembered undertaking a physics research project with the university. When asked what experiences they remembered from the project, the 11 open responses provided (3 students did not answer this or the following question) could be categorised as concerning the underlying science

> "[It] *helped us to really solidify our understanding of harmonics.*" (Student 2, cohort 1)
>
> "*Learning about how the magnetosphere works and its importance.*" (Student 8, cohort 2)

> "[I] *learned about exoplanets.*" (Student 14, cohort 3)

the process of undertaking university/research style work

> "*It was very intriguing to have played with actual data and have an attempt at analysing the sound wave forms that we were given.*" (Student 2, cohort 1)
>
> "*Setting up the experiments and working through problems as they arose.*" (Student 7, cohort 2)

> "*Working with other students on a project we did not know much about before and presenting it in front of a lot of people.*" (Student 11, cohort 3)
>
> "*Creating a formula to use in order to calculate [the] surface area of a scintillator which is hit directly by muons,*



*[and] observing building schematics to find how much matter muons pass through during travel into* [the] *building.*" (Student 12, cohort 3)

"*I remember getting to experience some more advanced practical physics that was more reminiscent of university lab work than school lab work.*" (Student 13, cohort 3)

the skills they developed through their involvement

"[I] *learned how to analyse sound in audacity.*" (Student 5, cohort 2)

"*I built good teamworking and project management skills*" (Student 11, cohort 3)

"*Learning to code in Python.*" (Student 14, cohort 3)

as well as having enjoyed the overall experience

"*having a great time and meeting some lovely people*" (Student 13, cohort 3)

When asked how they have used the experiences since, if at all, of the same 11 respondents all bar two provided examples. These concerned their skills development

"[It] *really helped to develop our teamwork skills, which I have used frequently in most things that I do in my academic education. Also there is a huge element problem-solving and how to undertake the project/study is fundamental in my Engineering degree for electronics, I have used it a lot.*" (Student 2, cohort 1)

"*Now I* [have] *used python in my computational physics module at university.*" (Student 3, cohort 1)

"*I am currently at university doing many group projects. Taking part in the Queen Mary magnetosphere project*
*has helped me improve my team building and communication skills.*" (Student 8, cohort 2)

"*The presentation helped me improve my public speaking and speaking confidence.*" (Student 13, cohort 3)

as well as activities they have done since

"*It was a good introduction to conducting experiments. I carried out a CREST gold research project after this experience.*" (Student 7, cohort 2)

"*I've tried to be confident to defend a project in front of people and to show an inquisitive attitude.*" (Student 11, cohort 3)

"*I now study architecture, so the observing building schematics and 3D mathematics were both useful experiences.*" (Student 12, cohort 3)

"*The lab work was useful as it gave me an idea of how to work in a uni lab, which is particularly helpful now that*
*I am at uni.*" (Student 13, cohort 3)

Of the two negative responses, both stating "*I haven't*", one caveated though that "*but that's just because I haven't had to do a group project since*" (Student 9, cohort 3). Overall, students' responses suggest that their PRiSE projects were lasting and beneficial experiences that they have been able to draw from in their subsequent educational activities and development.



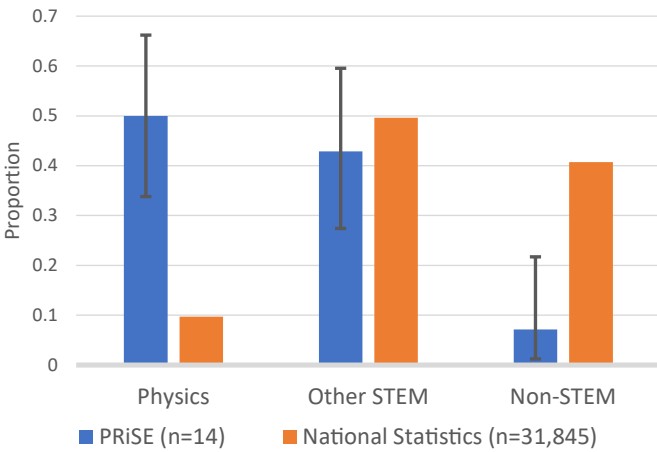

**Figure 4.** Degree destinations of PRiSE students (blue) compared to UK national statistics of A-Level physics students (orange). Error bars denote the standard Clopper and Pearson (1934) intervals.

### 4.2.2 Destinations

All the PRiSE students reported that they were studying at university when the survey was conducted, apart from one who due to their age (they were 14–15 when involved in PRiSE) intended to. We asked the students what subject they were studying at university (or planned to study in the case of the one student) giving the options of physics (or combination including physics), another STEM subject, or a non-STEM degree. The results of this are shown in blue in Figure 4, where we compare these to the degree destinations of physics A-Level students nationally (McWhinnie, 2012) in orange (this data only includes

students that went on to higher education). The 7 out of 14 PRiSE students going on to study a physics degree is considerably higher than the national rate of $9.7\%$ ($p = 1 \times 10^{-4}$ in a binomial test). While the number of students going on to study other STEM subjects (6 out of 14) are consistent with the national statistics, the increased uptake of physics leads to the overall STEM-degree proportion (13 out of 14) also being significantly greater than found nationally ($59.3\%$, $p = 0.012$). Given the diversity of schools involved (discussed in-depth in M.O. Archer, 2020) and the known barriers to STEM higher education

for underrepresented groups (e.g. Campaign for Science and Engineering, 2014; Hamlyn et al., 2020), it is highly unlikely these results can be explained simply by PRiSE schools tending to produce more physics and STEM students anyway. Another consideration may be that PRiSE students were already highly likely to continue their physics education beforehand anyway. While this did not appear to be the case for the 2019/20 cohort discussed in section 4.1.3, with it being shown that PRiSE led to some increased physics and STEM aspirations, the destinations results here came from different cohorts so this remains

a possibility. However, given that PRiSE has been a consistent framework (both in terms of schools targeting and delivery) throughout, it is reasonable to assume that the 2019/20 cohort is representative of the others used here and thus may be used as comparable samples. This then suggests that students' involvement in PRiSE may very well make them more likely to pursue physics and STEM degrees.





The students' reasons behind choosing their degree subjects and what influenced them varied. We note that one student out
of the 12 that responded to these questions referenced the research project (which was not prompted in the question)

> "*I did the sounds of space project that you organised a couple of years ago and am now pursuing a physics degree*
> *from Cambridge. Thanks for helping me find my enthusiasm for physics!*" (Student 4, cohort 1)

However, in general the responses were quite brief and did not give much insight into the likely many factors which may
have played a role in their subject choices (cf. L. Archer et al., 2013, 2020b). While for cohorts 2 and 3 we added questions
that explicitly asked about PRiSE's influence on the students' degree choices, the small number of responses and brief answers
simply highlight the need for more in-depth qualitative longitudinal evaluation, though this is challenging to undertake. We note
that there is little research into engagement programmes' effectiveness at influencing students' destinations, both correlative
and causal, as raised in a recent review (Robinson and Salvestrini, 2020). However, given that PRiSE shows some statistically
significant positive effects on students' aspirations, which are known to be highly resilient, at the 6-month stage as well as
PRiSE students being more likely to study physics and STEM degrees than would otherwise be expected (further statistically
significant effects), the results of this empirical enquiry into PRiSE's potential effect on students' destinations are perhaps
promising.

### 4.3 Co-production of research

Co-producing publishable physics research between researchers and school students is not an explicit aim of PRiSE, unlike for
instance the more researcher-driven ORBYTS programme (Sousa-Silva et al., 2018). The rationale behind this is explored fur-
ther in M.O. Archer et al. (2020). Nonetheless, in a few instances genuinely novel preliminary results have come from students'
independently motivated work on the MUSICS project (the other PRiSE projects are unlikely to result in publishable physics
research). The first of these originated from the 2016/17 cohort, where a group from a girls' school in an area of particularly
high deprivation discovered a series of decreasing-pitch "whistle" sounds which lasted several seconds (corresponding to 5
days in reality). Through collaboration with the students and further investigation by professional scientists, it was discovered
that these unexpected sounds corresponded to the natural oscillations of Earth's magnetic field lines following a solar storm.
Such wave events had been deemed rare previously but, due to the accessibility of exploring the sonified data, were found to in
fact be quite common thanks to the students' discovery. The work was presented at several international scientific conferences
and eventually published in the journal 'Space Weather' with the students and their teacher listed as co-authors (M.O. Archer
et al., 2018). Information about these developments resulting from the students' project work was continually passed on via
their teacher, who in turn responded with the students' comments:

> "*It was a very rewarding experience which allowed us an insight into the research conducted at university level.*
> *This helped us to develop crucial skills needed in the next years of our studies. It was truly amazing to hear how*
> *significant the event we found was and that it will be forming the basis of a proper scientific paper.*"
> "*Being a part of the university's research and the subsequent paper published is truly an amazing opportunity. It*





*was really interesting to find such a significant event and we gained so much experience and developed many skills during our research that will be useful in our university careers.*"

The publication garnered widespread media attention, for example featuring on BBC Radio 4's 'Inside Science'. Unfortunately we were unable to find out how the news of the publication had been shared across the school involved and affected other students' thoughts about physics. However, through publicising the result across all schools involved in PRiSE via teachers, it appears to have had a powerful effect on PRiSE students at other schools:

"*Hearing that other kids at other schools have actually produced a paper, it just gives you hope that it's actually something I can do.*" (Student, Summer Heights High, MUSICS, BBC Radio 4 interview, Oct 2018),

potentially highlighting through demonstration by their peers that (research-level) physics is something which is accessible to 'people like me', thereby breaking down known barriers to participation in physics and science generally.

While no other publications have resulted as of yet, a group of students (Imperial Academy) in 2018/19 identified undocumented instrumental noise present in the data which researchers and satellite operators were unaware of. Another group (Sunnydale High School) in 2019/20 decided to investigate the relationship between the recently discovered aurora-like STEVE (Strong Thermal Emission Velocity Enhancement) phenomenon (MacDonald et al., 2018) and magnetospheric ULF waves, finding some differences in wave activity during published STEVE events to typical levels, which might lead to promising results with significant additional work. Finally two more groups in 2019/20 (Morningwood Academy; and the partnership of Quirm College for Young Ladies, Sky High, Sycamore Secondary School, and Welton Academy) working on a follow-up campaign to the previous students' co-authored paper uncovered other wave events during solar storms with novel features that are currently being investigated further by professional scientists and may be publishable in the future.

## 5 Impact on teachers and schools

Possible impacts upon teachers and schools from PRiSE were first explored using qualitative responses to open-ended questions and then further investigated with quantitative data gathered in 2019-2020.

### 5.1 Thematic analysis

From a thematic analysis of all the qualitative data from open-ended questions collected across four years (2015–2018) from 21 teachers, we identified eight distinct areas (indicated in bold) in which teachers and schools seem to have been positively affected by their involvement in PRiSE. These codes have subsequently been applied to responses across all years of data (2015–2020, 45 teachers). Expressions of negative impact were rare, with only one teacher (Teacher 15, Tree Hill High School, MUSICS 2018) noting that the project had caused them "*a small amount of extra stress*", nonetheless, this teacher continued to engage with the programme in subsequent years. The first theme identified related to teachers **gaining new physics knowledge** based around the content of the projects:



> "*[It] added to* [my] *knowledge of standing waves giving more real-life examples of waves.*" (Teacher 8, Coal Hill School, MUSICS 2017)
>
> "*[It] introduced me to an area of physics where I have little experience. I have yet to teach the particles side of A-level physics. However, this project and the knowledge accumulated will be valuable when I do.*" (Teacher 20, Hogwarts, SCREAM 2018)

In turn, this has made them **more confident in discussing research** with students in general:

> "*[It has] given me confidence to explore physics beyond my areas of expertise / beyond the school specs.*" (Teacher 15, Spence Academy for Young Ladies, MUSICS 2018)
>
> "*It has re-ignited my interest in current research, and reminded me that complicated, cutting edge research can be more accessible than I sometimes think!*" (Teacher 36, Starfleet Academy, MUSICS 2020)
>
> "*[I have] re-engaged with research and research methods.*" (Teacher 41, Xavier's Institute for Higher Learning, SCREAM 2020)

Elements of the research projects have also been **implemented or referred to in teachers' regular lessons**:

> "*[I have been] able to use the detector with classes when teaching Year 12 particles.*" (Teacher 13, St Trinians, SCREAM 2016)
>
> "*It has been referred to* [in lessons] *in terms of what scientists do and the research process.*" (Teacher 15, Spence Academy for Young Ladies, MUSICS 2018)
>
> "*[It has given me] context when talking about Earth's magnetic field* [in lessons]." (Teacher 16, Tree Hill High School, MUSICS 2018)
>
> "*It has consolidated my understanding and teaching of exoplanets. I used some of the techniques in teaching detection of exoplanets in the astro topic of AQA's A-level going beyond the syllabus.*" (Teacher 19, Boston Bay College, PHwP 2018)

Another theme concerned teachers gaining **confidence in mentoring and/or supporting extra curricular activities**, such as developing their "*patience* [and] *encouragement*" (Teacher 6, Hogwarts, SCREAM 2016):

> "*[I have developed in] motivating students to attempt challenging problems.*" (Teacher 11, Prufrock Preparatory School , SCREAM 2017)

They also report developing a **variety of other skills** including algebra, data analysis, reviewing academic posters, using software such as Audacity and Excel, and computer programming:

> "*I have also enjoyed the personal challenge to my own coding abilities.*" (Teacher 39, Bending State College, PHwP 2020)

**Students' project work has been shared across the schools** such as via assemblies, displaying students' posters in classrooms or halls within the school, and publishing news stories on the school's website or in local papers (references not included so





as to preserve the anonymity of schools, teachers, and students). We note though that this was communicated to us informally either via email or during subsequent school visits rather than through the paper questionnaire. Nonetheless it seemed of

sufficient note to include as a theme. Following on from this, some teachers report in the survey that their school's involvement with the project has **raised the profile of physics or STEM within their school**:

> "[Students in lessons] *were impressed to hear of our 'muon project' and knowing we were involved with a university physics department helped them engage with us. If they think they and their teachers can be involved in research they are more motivated.*" (Teacher 1, Hogwarts, SCREAM 2015)

> "[It] *gave prestige to the Physics department at the college.*" (Teacher 9, Xavier's Institute for Higher Learning, MUSICS 2017)

Finally, some teachers feel that they and their schools have **developed a relationship with the university**:

> "[It] *created a link to HE.*" (Teacher 15, Spence Academy for Young Ladies, MUSICS 2018)
>
> "*We feel involved in a very interesting* [research] *project.*" (Teacher 27, Hogwarts, SCREAM 2019)

This is further backed-up by the significant repeated buy-in of teachers and schools after completing the project, with $70 \pm 10\%$ returning for multiple years of PRiSE projects, further explored in M.O. Archer (2020), thereby changing how they interact with universities (e.g. not just attending one-off events) and solidifying the above mentioned impacts:

> "*I am more confident in my second year.*" (Teacher 21, Hogwarts, SCREAM 2018)
>
> "*Now I've done one project I feel better equipped to get things going myself.*" (Teacher 39, Bending State College,

> PHwP 2020)

We use these eight areas of impact on teachers and schools for subsequent quantitative analysis in the next section. However, we note with further teacher survey responses in 2019–2020 we have identified an additional theme. This pertains to **teachers' preconceptions of their students' ability**:

> "*I am now more aware of what our students are capable of - not just listening to visiting speakers but being actively

> engaged in real-world research!*" (Teacher 10, Prufrock Preparatory School, SCREAM 2017)
>
> "[It has] *made me more enthusiastic to engage students in real research.*" (Teacher 17, Sunnydale High School, MUSICS 2018)
>
> "*The project allowed me to identify students that were genuinely interested and committed to Physics. It also gave me evidence that my students should study science further at university. I was able to pass this on to parents and

> universities.*" (Teacher 31, Quirm College for Young Ladies [and partner schools], MUSICS 2020)
>
> "*It has been inspiring to see my students self-organising so well together.*" (Teacher 43, Sunnydale High School, MUSICS 2020)





## 5.2 Quantitative analysis

Based on the areas of impact on teachers and schools emerging from the qualitative data (from 2015–2018), from 2019 onwards
we sought to quantitatively assess how prevalent they might be. Teachers ($n = 23$) were asked to identify for each of the 8 themes whether they felt that they (or their school) had been affected by the project in that area, using the closed options of: "I have", "I will eventually", "I have not", and "Unsure". This scale was chosen over a 5-point Likert due to an expected low level of responses. We exclude any blank or unsure answers (which were rare) and divide the remaining 172 responses across the 8 themes into negatives ("I have not") and positives, with the latter being subdivided into planned ("I will") and definite ("I
have") impacts.

Figure 5a shows the distributions of these results for each impact area along with the overall results obtained from totalling all responses. We find that all areas, apart from mentoring ($p = 0.115$), have statistically significant positive majorities in binomial tests ($p < 0.03$). Only learning new physics and developing a relationship with QMUL have majority definite responses ($p < 0.011$). Coding the responses to values of 1 (negative) to 3 (definite) the overall average is $2.52 \pm 0.06$, which is greater than 2
to a very high level of confidence. In fact all categories, apart from mentoring ($p = 0.108$) and sharing outcomes ($p = 0.119$), are statistically significant in one-sample Wilcoxon signed-rank tests against null hypotheses of 2. However, all categories' distributions are consistent with the overall results in Wilcoxon rank sum tests, thus while there are differences between the distributions such as slightly more teachers feeling they have not developed mentoring skills or having not yet shared their students' work across their schools (understandable given the question was posed at the student conference) these variations
are moderate and not statistically significant.

Figure 5b shows the distribution of the number of positive impacts claimed per teacher. No teachers responded negatively in all categories and thus all seem to have been positively affected in some way. The average number of positive categories indicated by teachers was $6.2 \pm 0.4$ out of 8. The bivariate distribution of definite impacts per teacher as a function of the number of positive impacts they indicated is shown inset, revealing teachers tended (apart from in one case) to indicate the
majority of positive impacts claimed had already occurred. The average number of definite impacts was $5.2 \pm 0.5$. Out of the 23 teachers, only 5 gave the same answer in all categories (in these cases all definite responses) suggesting the results are largely reliable and likely did not fall prey to unreflective responses. Therefore, it appears that the identified areas of impact upon teachers and schools as a result of PRiSE may indeed be quite widespread.

## 6 Conclusions

We have investigated the medium- and long-term impacts on students, teachers, and schools who have participated in a 6-month-long programme of physics 'research in schools' projects, open-ended investigations for school students based around cutting-edge STEM research. This programme, 'Physics Research in School Environments' (PRiSE), has involved a diverse range of London schools and we have used evaluation data captured from questionnaires across its entire 6 year duration to date.





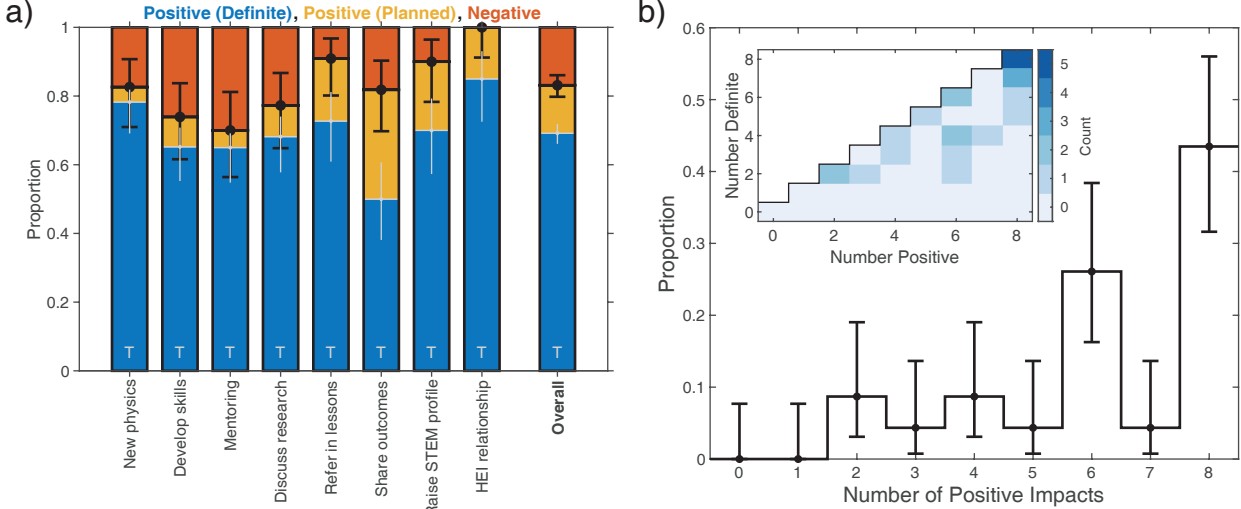

**Figure 5.** Quantitative results of impacts on teachers and schools ($n = 23$). a) Distribution of teachers' responses for each impact category. Results are divided (black lines and associated error bars) into negative (red) and positive responses, with the latter subdivided (grey lines and error bars) into 'definite' (blue) and 'planned' (yellow). b) Distribution of the number of positive impact categories reported per teacher, with the bivariate distribution of the number of definite impacts vs. positive ones shown inset. Error bars denote standard Clopper and Pearson (1934) intervals.

Medium-term impacts on the participating 14–18 year-old school students were assessed after they had completed their 6-month-long projects. Students' confidence in relevant scientific topics and methods seems to have substantially increased as a result of PRiSE, with nearly all students reporting this benefit. Furthermore, through experiencing and being involved in research-level physics, students report having gained new, or further developed existing, skills. Both of these impacts upon students have been corroborated by teachers' observations. While the students involved with PRiSE were fairly committed to STEM in general beforehand, our data suggest that they had no clear bias in aspirations towards the subject of physics in particular. Following the programme it appears that students' attitudes towards pursuing STEM were typically maintained or confirmed through their involvement and physics aspirations seem to have been moderately enhanced. We find no evidence that these impacts varied by the different projects or schools. These results should be deemed successful, as a drop-off in STEM aspirations is often seen at this age (Davenport et al., 2020), with these issues being particularly pertinent in physics (L. Archer et al., 2020a). Thus at this stage of a student's educational journey they are likely to require interventions that sustain and support their science identity which, in turn, has an influence on their educational choices (L. Archer and DeWitt, 2016).

Longitudinal evaluation has also been performed for three cohorts of PRiSE students 3 years after they commenced their projects. While a relatively small sample, the evidence suggests that these projects have been highly memorable and beneficial experiences that students have been able to draw upon in their later educational activities and development. The data on PRiSE students' degree destinations show increased uptake of physics and STEM at degree-level than would typically be expected,



suggesting that their involvement in the research projects has helped transform their aspirations into destinations — a key aim of the programme. Further in-depth qualitative research, such as interviews or focus groups, could provide richer and more reflective information on how students' thoughts and feelings about their association with physics and STEM may have been affected by participating in PRiSE, given the nuance and multiple factors at play with students' aspirations in general, which

are difficult to capture and interpret with questionnaire data alone (L. Archer et al., 2013, 2020b).

The impacts upon students reported in this paper only relate to those who completed the 6-month programme. However, as should be expected for any extended programme, there is some drop-off in participation with PRiSE. This has been explored in more detail in M.O. Archer (2020), demonstrating that students' success with PRiSE appears to be independent of background and thus not clearly patterned by societal biases present within the field (e.g. Campaign for Science and Engineering, 2014).

Currently though we have no data on what impact the programme has on those students who drop out. While no teachers have communicated any negative effects on students who do not continue, with some highlighting informally their students' attitudes towards the projects, this requires further formal investigation. Such work is required to ensure that no negative effects are being felt by these students and potentially discover what positives, if any, may result from even partial participation. Furthermore, we have no evidence that the PRiSE approach would be effective for students who are generally uninterested or

unengaged with STEM. Indeed, it seems unlikely that such students would want to persist with an extended and challenging extra-curricular physics programme. Young people's aspirations towards science begin to form at an early age (L. Archer et al., 2013, 2020b) and therefore interventions throughout their educational journey need to align with their needs and wants, from initial inspiration and positive associations, to informing on career-focused aspects, and finally sustaining those built aspirations (Davenport et al., 2020). PRiSE only aims to address that final part of the chain, since no single programme can fit all stages.

Evaluation of the impacts on teachers and schools has identified several themes. By collaborating on PRiSE, teachers can gain new physics knowledge, become more confident in discussing research, and integrate aspects of the research projects into their regular lessons. Teachers also report developing various technical skills, gaining confidence in mentoring, and reassessing their preconceptions of students' potential. While all these positive changes to teachers' practice will likely be felt across their wider schools, there is more direct evidence of the school environment being affected such as through students' project

work being championed, the profile of physics or science being raised, and a university–school relationship being established with significant repeated buy-in from schools over several years. These impacts appear to be fairly widespread across the teachers and schools involved in PRiSE. We note that these results share many similarities to those reported by Rushton and Reiss (2019) for IRIS (2020) from interviews with 17 teachers. The PRiSE programme features much greater diversity in schools, with an over-representation of disadvantaged groups in many metrics considered (M.O. Archer, 2020). Furthermore,

as discussed in M.O. Archer et al. (2020), PRiSE does not rely on such a teacher-driven model instead providing a wealth of resources and interventions to support teachers' and schools' participation. The similar impacts thus highlight that, with the right support, teachers and schools from a variety of contexts can benefit from 'research in schools' projects. Further research could investigate the validity of the participating teachers' remarks of the impact on the schools' environments, for instance through in-depth interviews or focus groups with other teachers in the schools.





The impacts upon participating students, teachers, and schools discussed in this paper show real promise for the emerging field of 'research in schools' initiatives. They suggest that with more similarly designed and supported programmes at other institutions, we may be able to start to address a key part of the chain of the wider issue of uptake and diversity not just in physics but potentially STEM also. We stress, however, that multi-faceted approaches from a variety of different stakeholders and organisations are required to implement real change on this entire issue, but 'research in schools' may be able to form one

piece of the puzzle.

**Appendix A: 6-month stage evaluation questions**

Here we list the questions considered within this paper posed in the PRiSE-wide questionnaires at the 6-month stage evaluation. We detail the phrasing used, how participants could respond, and which years the question was asked. Follow-on questions are indicated by indentation and a down-right arrow (↳). The following questions were posed to students:

| Question | Response type | Year(s) |
| --- | --- | --- |
| In what way has this project affected you | Open Text | 2016–2020 |
| What skills, if any, has the project helped you develop | Open Keywords | 2016–2020 |
| How has doing the project affected your thoughts about future subject choices / careers | Open Text | 2018–2019 |
| Before working on the project, how likely were you to continue with the following (Physics/STEM) in the future | 5-point Likert | 2020 |
| ↳After working on the project, how likely are you now to continue with the following (Physics/STEM) in the future | 5-point Likert | 2020 |
| ↳How has working on the project affected your thoughts about these future subject choices (Physics/STEM) | 5-point Likert | 2020 |
| ↳Please explain how or why? | Open Text | 2020 |

The questions asked of teachers were:

| Question | Response type | Year(s) |
| --- | --- | --- |
| In what way has this project affected your students | Open Text | 2015–2020 |
| In what way has this project affected you | Open Text | 2015–2018 |
| What skills, if any, has the project helped your students develop | Open Keywords | 2016–2020 |
| What skills, if any, has the project helped you develop | Open Text | 2016–2018 |
| Have you found the project useful in your lessons | 5-point Likert | 2015–2018 |
| ↳Please tell us why / why not | Open Text | 2015–2018 |




| | | | |
|---|---|---|---|
| Do you feel you have been affected by the project in any of the following ways (8 categories) | Closed Options | 2019–2020 | |
| ↳Are there any other ways you feel you've been affected | Open Text | 2019–2020 | |

## Appendix B: 3-year stage evaluation questions

The following questions were asked of students in the 3-year stage evaluation via an online form.

| Section | Question | Response type | Required | Go to section | Cohort(s) |
|---|---|---|---|---|---|
| 1 | What school year would you have been in during the academic year [cohort year] | Closed Options | Y | | 1–3 |
| | Do you remember undertaking a Queen Mary Physics & Astronomy research project that year | Yes/No | Y | Yes: 2<br>No: 3 | 1–3 |
| 2 | What experiences do you remember from the project | Open Text | N | | 1–3 |
| | How have you used those experiences since, if at all | Open Text | N | 3 | 1–3 |
| 3 | Are you studying at a Higher Education Institution (HEI) e.g. a University | Closed Options | Y | Yes: 4<br>Intend to: 5<br>No: Submit | 1–3 |
| 4 | Which HEI are you studying at | Open Text | N | 5 | 1–3 |
| 5 | What subject are you studying or plan to study at a HEI | Closed Options | Y | | 1–3 |
| | What are/were your reasons for studying that subject | Open Text | N | | 1–3 |
| | What influenced you to study that subject | Open Text | N | 6 | 1–3 |
| 6 | Did the research project in any way influence your subject choice | Closed Options | Y | | 2–3 |
| | How would you say the research project affected your subject choice (if applicable) | Open Text | N | Submit | 2–3 |

*Data availability.* Data supporting the findings of this study that is not already contained within the article or derived from listed public domain resources are available on request from the corresponding author. This data is not publicly available due to ethical restrictions based
on the nature of this work.



*Author contributions.* MOA conceived the programme and its evaluation, performed the analysis, and wrote the paper. JDW contributed towards the analysis, validation, and writing.

*Competing interests.* The authors declare that they have no conflict of interest.

*Acknowledgements.* We thank Dominic Galliano, Olivia Keenan, and Charlotte Thorley for helpful discussions. MOA is grateful for funding
from the Ogden Trust. This programme has been supported by a QMUL Centre for Public Engagement Large Award, and STFC Public Engagement Small Award ST/N005457/1.



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
