# Peer review of "“Thanks for helping me find my enthusiasm for physics!” The lasting impacts ‘research in schools’ projects can have on students, teachers, and schools"

_Geoscience Communication, 2020_

## Referee Comment (RC1) · Anonymous Referee #1 · 16 Oct 2020

This paper is very interesting and it presents, in a very well organised and clear manner, the impact of a "Research in Schools" programme on participating students, teachers and their schools. The study was well designed and the authors draw very honest conclusions from the acquired data and from its analysis, in particular keeping the conclusions specific to the cohorts that participated in the programme. While nontrivial, the inclusion of control groups in the design would have added an extra depth to the study. Perhaps that will be possible in future studies.

Regarding the results presented on section 4.1.2 Skills, I am left wondering if students

may have been influenced by teachers and/or by introductory materials handed out to them at the start of their participation in PRiSE, in building the list of developed skills. Was a list of "skills to be developed. . ." provided to them in order to "sell" the project? I suggest to include a comment on that.

On section 6 Conclusions, on line 538, the authors state ". . .suggest that these projects have been highly memorable. . .". I agree that the data fully supports that the projects were memorable, but the classification as highly memorable seem a more subjective opinion by the authors. I suggest leaving out "highly" on that sentence.

The citation included in the title, while making it catchier, introduces some unnecessary bias. A different citation could have been, e.g., "I never really saw Physics as a choice for me . . . and the project hasn't changed my mind about this." Perhaps it is better to leave any citation out of the title. Having said that, I thoroughly recognise that the authors' choice of citation is a much better reflection of the observed overall trend than the example I use above.

I cannot finish without stressing the importance of this and of similar studies. They are crucial to better understand the real impact of science communication activities carried out by HE institutions, research centres, science centres and museums, and to guide institutions and individuals into better practices.

P.S. The sentence beginning towards the end of line 244 reads "A similar proportion to with physics of 68%...". Being a non-native English speaker, I may be missing some nuance of the English language, but that sentence doesn't sound right to me. Apologies if it is my fault.

---

## Referee Comment (RC2) · Anonymous Referee #2 · 3 Nov 2020

This is an excellent paper, and describes important, and thorough research into the effects of an engagement programme - something which is notoriously hard to do. My comments are largely very minor

Section 3: In point 5, the authors mention reliability. It would be interesting to know how reliable they found the codes (i.e. what were different between the first and second coder, and was it signifiant)

4.1.1: (sentence 2) "Additionally they were asked to reassess their confidence before

[Figure]

having undertaken the project." I found this ambiguous - reassessment implies a second assessment, when the text mentions that there was no pre-assessment. Perhaps "retrospectively assess" would be more accurate?

Figure 1: I very much like the figures in the article, and they are all clear. With this figure, it may be worth considering applying some transparency to the points as some are overlapping. I am, however, willing to believe that this makes it too confusing, but it is something the authors should consider.

Table 4: (caption) if 11 weren't placed in dimensions, it may be clearer to say using n=52 of 63 responses?

Figure 4: what intervals are the error bars (1-sigma, 95%?). It is worth noting that these intervals are not reliable with small n or low probabilities of success. I suggest at least an acknowledgement that these should be treated at indicative given the sample size.

Section 4.3: in the final paragraph there are details of (as yet) unpublished works which name the "anonymised" schools. Does this risk de-anonymising the schools used here, if the results are published in the future?

Section 5.2: The authors use a null hypothesis of 2. Would a better quantitative test to simply be to code positive vs negative, without the division into planned and definite? (i.e. give definite a score of 2 as well for this purpose, with a null hypothesis of 1.5)? At present the null hypothesis is that 2/3 of respondents claim a positive impact.

Figure 5: I found the grey error bars hard to spot, as they are narrow and overlap the black error bars. Perhaps thicker lines and/or offset horizontally with respect to the black error bars?

---

## Author Comment (AC1) · 22 Nov 2020

**This paper is very interesting and it presents, in a very well organised and clear manner, the impact of a "Research in Schools" programme on participating students, teachers and their schools. The study was well designed and the authors draw very honest conclusions from the acquired data and from its analysis, in particular keeping the conclusions specific to the cohorts that participated in the programme. While nontrivial, the inclusion of control groups in the design would have added an extra depth to the study. Perhaps that will be possible in**

[Figure]

**future studies.**

We thank the reviewer for their time in assessing the manuscript. In our companion paper, M.O. Archer et al. (2020), we discussed the ethical reasoning behind a lack of control groups in our evaluation of the programme. We will mention this limitation to the study and add a reference to the reasoning in the manuscript as follows:

> No control groups were established due to ethical considerations, further explored in M.O. Archer et al. (2020), which slightly limits this impact study. However, where possible we draw from publicly available benchmark data.

Archer, M. O., DeWitt, J., and Thorley, C.: Transforming school students' aspirations into destinations through extended interaction with cutting-edge research: Physics Research in School Environments, Geosci. Commun. Discuss., https://doi.org/10.5194/gc-2020-35, in review, 2020.

**Regarding the results presented on section 4.1.2 Skills, I am left wondering if students may have been influenced by teachers and/or by introductory materials handed out to them at the start of their participation in PRiSE, in building the list of developed skills. Was a list of "skills to be developed. . ." provided to them in order to "sell" the project? I suggest to include a comment on that.**

The reviewer raises an interesting point. None of the resources provided with the projects indicated expected skills to be developed, however, we cannot comment on what teachers may have said to their students outside of interventions. We will make the following addition:

> Skills development was not mentioned in any of the PRiSE projects' resources (e.g. there was no list of "skills to be developed") so we are confident in the validity of these results, though we cannot rule out that teachers may have influenced students' answers.
On section 6 Conclusions, on line 538, the authors state ". . .suggest that these projects have been highly memorable. . .". I agree that the data fully supports that the projects were memorable, but the classification as highly memorable seem a more subjective opinion by the authors. I suggest leaving out "highly" on that sentence.

We will make this change.

The citation included in the title, while making it catchier, introduces some un-necessary bias. A different citation could have been, e.g., "I never really saw Physics as a choice for me . . . and the project hasn't changed my mind about this." Perhaps it is better to leave any citation out of the title. Having said that, I thoroughly recognise that the authors' choice of citation is a much better reflec-tion of the observed overall trend than the example I use above.

The reviewer is correct that the quote used in the title was designed to make it catchier. We believe the quote exemplifies the overall quantitative and qualitative results of the paper, as the reviewer points out, that PRiSE students' physics aspirations (after 6 months) and university destinations (after 3 years) were enhanced and that physics teachers' practice was developed. Therefore, we feel its usage is appropriate in sum-marising the paper's conclusions.

I cannot finish without stressing the importance of this and of similar studies. They are crucial to better understand the real impact of science communication activities carried out by HE institutions, research centres, science centres and museums, and to guide institutions and individuals into better practices.

We thank the reviewer for this comment.

P.S. The sentence beginning towards the end of line 244 reads "A similar pro-portion to with physics of 68%...". Being a non-native English speaker, I may be missing some nuance of the English language, but that sentence doesn't sound

**right to me. Apologies if it is my fault.**

We will adjust this sentence to make it clearer as follows

> $68\pm9\%$ of students indicated PRiSE's likely positive influence on their STEM aspirations (a result similar to physics aspirations).

---

## Author Comment (AC2) · 22 Nov 2020

Martin O. Archer and Jennifer DeWitt

m.archer10@imperial.ac.uk

**This is an excellent paper, and describes important, and thorough research into the effects of an engagement programme - something which is notoriously hard to do. My comments are largely very minor.**

We thank the reviewer for their time and comments.

**Section 3: In point 5, the authors mention reliability. It would be interesting to know how reliable they found the codes (i.e. what were different between the**

[Figure]

**first and second coder, and was it significant)**

Overall there was 92% agreement between the two coders, which corresponds to a Cohen's kappa of 0.836 (Cohen's kappa is unity minus the ratio of observed disagreement to that expected by chance, hence ranges from 0 to 1). Disagreements were resolved by discussion to arrive at the final coding presented in the paper. We will add these points to the paper.

**4.1.1: (sentence 2) "Additionally they were asked to reassess their confidence before having undertaken the project." I found this ambiguous - reassessment implies a second assessment, when the text mentions that there was no pre-assessment. Perhaps "retrospectively assess" would be more accurate?**

We thank the reviewer for this suggested wording, which we will now use.

**Figure 1: I very much like the figures in the article, and they are all clear. With this figure, it may be worth considering applying some transparency to the points as some are overlapping. I am, however, willing to believe that this makes it too confusing, but it is something the authors should consider.**

We have tried the reviewer's suggestion, but found it makes things less clear. The main point of Figure 1 is that almost all the points lie in the upper triangle, indicating a positive effect, which is clear. The precise locations of each datapoint are not so crucial in this context.

**Table 4: (caption) if 11 weren't placed in dimensions, it may be clearer to say using n=52 of 63 responses?**

We thank the reviewer for this suggestion, which we have adopted.

**Figure 4: what intervals are the error bars (1-sigma, 95%?). It is worth noting that these intervals are not reliable with small n or low probabilities of success. I suggest at least an acknowledgement that these should be treated at indicative given the sample size.**

The error bars represent the standard (1 sigma) confidence interval using the Clopper and Pearson method. This was mentioned in the caption and discussed further in the methods (lines 111-114), explaining that they are a conservative estimate based on the exact binomial distribution. Therefore, the error bars do not rely on the normal approximation, which is known to be unreliable for small n or low probabilities. We will further clarify in the captions of figures that the word "standard" refers to 1 sigma.

**Section 4.3: in the final paragraph there are details of (as yet) unpublished works which name the "anonymised" schools. Does this risk de-anonymising the schools used here, if the results are published in the future?**

The reviewer raises a good point. The reason the schools' pseudonyms were added in this section were so readers could check the type of schools (i.e. independent or state, high Free School Meals etc.) to show that these outcomes were not biasing to privileged schools. However, to further protect anonymity we will comment on these aspects within the text rather than providing the pseudonyms.

**Section 5.2: The authors use a null hypothesis of 2. Would a better quantitative test to simply be to code positive vs negative, without the division into planned and definite? (i.e. give definite a score of 2 as well for this purpose, with a null hypothesis of 1.5)? At present the null hypothesis is that 2/3 of respondents claim a positive impact.**

The reviewer's suggested test is less strict than the one adopted in the paper that they refer to. The one-sample Wilcoxon signed-rank test tests whether the median is significantly different from a hypothetical value, as explained on lines 117-120. Applying the reviewer's suggested test gives unilaterally smaller p-values.

We also note that a test of positives vs. negatives was also already performed (lines 502-503). The reason behind a null hypothesis of 2 in the later tests was that the "I will" response might be construed by some as neutral, therefore potentially biasing the positive results. Our analysis has thus taken both interpretations into account. We will

add this reasoning to the paper as follows:

> We acknowledge some may consider the "I will" response as neutral and thus our analysis takes both interpretations into account.

**Figure 5: I found the grey error bars hard to spot, as they are narrow and overlap the black error bars. Perhaps thicker lines and/or offset horizontally with respect to the black error bars?**

We have made the grey error bars thicker.